# Small GTPases of the Ras and Rho Families Switch on/off Signaling Pathways in Neurodegenerative Diseases

**DOI:** 10.3390/ijms21176312

**Published:** 2020-08-31

**Authors:** Alazne Arrazola Sastre, Miriam Luque Montoro, Patricia Gálvez-Martín, Hadriano M Lacerda, Alejandro Lucia, Francisco Llavero, José Luis Zugaza

**Affiliations:** 1Achucarro Basque Center for Neuroscience, Science Park of the Universidad del País Vasco/Euskal Herriko Unibertsitatea (UPV/EHU), 48940 Leioa, Spain; alazne.arrazola@ehu.eus (A.A.S.); miriamluquem@gmail.com (M.L.M.); 2Department of Genetics, Physical Anthropology, and Animal Physiology, Faculty of Science and Technology, UPV/EHU, 48940 Leioa, Spain; 3Department of Pharmacy and Pharmaceutical Technology, Faculty of Pharmacy, University of Granada, 180041 Granada, Spain; galmafarma@gmail.com; 4R&D Human Health, Bioibérica S.A.U., 08950 Barcelona, Spain; 5Three R Labs, Science Park of the UPV/EHU, 48940 Leioa, Spain; hadrilac@gmail.com; 6Faculty of Sport Science, European University of Madrid, 28670 Madrid, Spain; alejandro.lucia@universidadeuropea.es; 7Research Institute of the Hospital 12 de Octubre (i+12), 28041 Madrid, Spain; 8IKERBASQUE, Basque Foundation for Science, 48013 Bilbao, Spain

**Keywords:** GTPases, neurodegeneration, Alzheimer, Parkinson, Ras, Rap, Rho, Rac, Cdc42

## Abstract

Small guanosine triphosphatases (GTPases) of the Ras superfamily are key regulators of many key cellular events such as proliferation, differentiation, cell cycle regulation, migration, or apoptosis. To control these biological responses, GTPases activity is regulated by guanine nucleotide exchange factors (GEFs), GTPase activating proteins (GAPs), and in some small GTPases also guanine nucleotide dissociation inhibitors (GDIs). Moreover, small GTPases transduce signals by their downstream effector molecules. Many studies demonstrate that small GTPases of the Ras family are involved in neurodegeneration processes. Here, in this review, we focus on the signaling pathways controlled by these small protein superfamilies that culminate in neurodegenerative pathologies, such as Alzheimer’s disease (AD) and Parkinson’s disease (PD). Specifically, we concentrate on the two most studied families of the Ras superfamily: the Ras and Rho families. We summarize the latest findings of small GTPases of the Ras and Rho families in neurodegeneration in order to highlight these small proteins as potential therapeutic targets capable of slowing down different neurodegenerative diseases.

## 1. Introduction

Eukaryotic cells permanently maintain communication with the extracellular medium through some molecules such as growth factors, hormones, peptides, and ions. These signaling mediators (agonists or antagonists) can bind to specific receptors, which promote the internalization of the message that spreads via signaling cascades in order to drive a specific cellular response [1].

The above-mentioned processes share common elements that are recruited in the presence of the different stimuli and have an active role in both several and different responses. One of those recruited elements are small guanosine triphosphatases (GTPases) of the Ras superfamily. Those proteins are key regulators of many important and different cellular events that take place in eukaryotic cells such as proliferation, differentiation, or apoptosis [2,3].

Small GTPases act as intracellular molecular switches and can be found in two states: the inactive state in which the GTPase is bound to guanosine-5′-diphosphate (GDP), and the active state where the GTPase is bound to guanosine-5′-triphosphate (GTP). Three main molecules regulate the activation/deactivation cycle: guanine nucleotide exchange factors (GEFs), GTPase-activating proteins (GAPs), and GDP-dissociation inhibitors (GDIs). The GEFs favor the release of the GDP, allowing the binding of GTP to the small GTPase, therefore regulating the transition from the inactive to the active state. On the other hand, GAPs are responsible for the inactivation of the small GTPase by activating its intrinsic GTPase activity, resulting in the hydrolysis of GTP generating GDP and inorganic phosphate (Pi). The last main regulator molecules are GDIs, which are responsible for the inhibition of the small GTPases. GDIs prevent dissociation of GDP bound to the GTPase, consequently, keeping the small GTPase sequestered and inactive in the cytosol. Thus, the process by which the small GTPases pass from the active to the inactive configuration and vice versa, together with the actions of GEFs, GAPs, and GDIs on GTPases, is called the GTPases cycle [4] (Figure 1).

Moreover, post-translational modifications can regulate GTPases activity. In this regard, C-terminal or N-terminal regions prenylations can modify GTPases, which permits the attachment to specific proteins as well as membranes [5]. In fact, they can be farnesylated, geranylgeranylated, or palmitoylated in the C-terminal region, covalently linked to cysteine, and myristoylated in the N-terminal region [5].

The small GTPases of the Ras superfamily are classified into five large families—Ras, Rho, Rab, Arf, and Ran—where each family is specialized in the regulation of specific functions. However, it has been described that GTPases from different families can cooperate with each other in many circumstances [6,7].

The Ras family is implicated in cellular growth control and metabolism [2,3], and it cooperates with the Rho family in the regulation of the cell cycle, gene expression, and cellular transformation [2,3]. The Rho family is specialized in actin cytoskeleton reorganization. The Rab family is responsible for the intracellular vesicle and membrane trafficking, whereas the Arf family regulates the formation of the vesicles and the intracellular transport [2,3]. Lastly, the Ran family controls the nucleocytoplasmic transport and microtubule organization [2,3].

Depending on the context (i.e., cell type or the stimulus), activated GTPases lead to cellular responses such as proliferation, differentiation, motility, survival, or apoptosis [2,3]. Taking into account the importance of these responses at a cellular physiology level, mutations or alterations in the Ras superfamily of GTPases have been associated with different diseases, such as cancer [8], as well as vascular [9] and neurodegenerative diseases [10].

Neurodegenerative diseases are characterized by the progressive loss of specific subsets of neurons [11]. The original trigger of neuronal death is still unknown, although aging is considered a risk factor for neurodegenerative diseases. Many of these disorders share an abnormal accumulation of misfolded peptides or proteins in the brain and spinal cord. These deposits of insoluble proteins could accumulate with time, and they could be more toxic when neurons are old, as they lose the ability to degrade such proteins. Apart from protein seeding and propagation, the pathological hallmarks behind a neurodegenerative disease include altered protein quality control, dysfunctional mitochondrial homeostasis, autophagy and lysosomal dysregulation, stress granules, synaptic toxicity, neuroinflammation due to dysregulated glial cells, and maladaptive innate immune responses [11]. All these features culminate in neuronal death.

Neurodegenerative diseases are classified depending on main clinical characteristic, anatomic distribution of the neuronal loss, or fundamental molecular alteration (e.g., the protein that is being accumulated, as well as its neuroanatomical distribution). Hence, depending on which protein is accumulated, neurodegenerative diseases can be classified in different groups: amyloidosis diseases, Tau protein impairments, and α-synucleinopathies that are characterized by the accumulation and aggregation of β-amyloid (Aβ) peptides, Tau protein impairments, and α-Synuclein (α-syn) accumulation, respectively [11].

The misfolded proteins mentioned above are hallmarks of the most prevalent (and, therefore, most studied) neurodegenerative diseases, such as Alzheimer’s disease (AD) and Parkinson’s disease (PD) [11]. AD and PD are the two most common neurodegenerative diseases.

The most common form of dementia, AD, is a progressive and irreversible neurodegenerative disorder [12]. It is characterized by the abnormal accumulation of both Tau protein and Aβ peptides. The aggregation of hyperphosphorylated Tau protein provokes the formation of neurofibrillary tangles (NFT) in the intracellular region [13]. NFT formation typically parallels neuronal loss and, therefore, it has been associated with AD severity. The accumulation and spreading of Aβ peptides (called senile plaques) are also a hallmark of AD. The accumulation and aggregation of both proteins induce the degeneration of hippocampal and cerebral cortex neurons, resulting in the typical AD symptoms that include memory loss, spatiotemporal disorientation, or behavioral changes among others [12].

PD is the second most common neurodegenerative disease [14]. It is characterized by intracellular inclusions containing α-syn aggregates forming the Lewy bodies (LB). Forms of PD are caused by different mutations in leucine rich-repeat kinase 2 (LRRK2), Parkin E3-ubiquitin ligase, and PTEN-induced putative kinase 1 (PINK1) [15,16]. PD cellular characteristics consist on the degeneration of dopaminergic (DA) neurons in *substantia nigra*, resulting in symptoms such as bradykinesia, resting tremor, muscular rigidity, postural instability, and dementia.

Furthermore, the accumulation of neurodegeneration-related proteins mentioned above is related to the alteration of the intracellular neuronal and glial pathways. All these pathological mechanisms could be due to dysregulated signal transduction pathways that alter neuron and glial functionality.

As previously mentioned, small GTPases are key molecules that integrate cellular inputs to elaborate a biological response. Due to their important functions, members of the Ras superfamily have been associated with AD and PD [10]. Thereby, small GTPases of the Ras superfamily have long been shown to be involved in AD [17]. In PD, small GTPases and/or missense mutations in GTPases are implicated since they participate in the signal transduction mediated by LRRK2, Parkin, and PINK1 [18]. Apart from these pathologies, small GTPases are also involved in many other neurodegenerative disorders, such as amyotrophic lateral sclerosis (ALS), Huntington’s disease (HD), and even Creutzfeldt–Jakob disease (CJD) [10].

Therefore, the dysregulation of specific GTPases of the Ras superfamily, or their regulators or effector molecules, leads to aberrant signaling pathways and/or pathological cell responses that could cause neurodegeneration in AD and PD. Understanding how these GTPase-mediated molecular events are dysregulated in neurodegenerative diseases can help us understand the cause of neurodegeneration. This might provide new avenues for the development of therapeutic approaches for neurodegenerative diseases.

Thus, in this review, we go deeper into the dysregulated/aberrant signaling pathways controlled by the Ras superfamily GTPases that culminate in neurodegeneration and further progression of two major neurodegenerative diseases (AD and PD). We specifically focused on the two most studied families of the Ras superfamily: Ras and Rho. We summarize the latest findings to highlight the role of small GTPases of the Ras and Rho families in neurodegenerative processes.

## 2. Small GTPases of the Ras Family

This family of GTPases regulates biological responses such as proliferation, differentiation, and migration [3]. However, it is classically associated to tumor development processes, therefore, being classified as proto-oncogenes [8].

### 2.1. Ras GTPase

Ras GTPases activation is promoted by GEFs such as SOS, RasGRF, RasGRP, and RalGDS, whereas Ras GTPases are inactivated by GAPs such as RASA 1–3, RASAL 1–3, DAB2IP, NF1.SPRED 1–3, and SYNGAP1 [19]. Apart from GEFs and GAPs, Ras activity can also be regulated by scaffold proteins such as KSR, which favors Ras signaling via MAP-kinase (MAPK), its main effector molecule [20].

Considering the importance of Ras-regulated responses, a dysregulation of this GTPase could underlie some neurodegenerative diseases. In fact, an abnormal activity of the Ras pathway has been associated with Alzheimer’s disease [21]. One of the first studies carried out in AD patients described that they expressed abnormally elevated levels of the adaptor protein growth factor receptor-bound protein 2 (Grb2) and SOS-1, a RasGEF [22]. Moreover, some evidence suggest that the MAPK pathway could be involved in Tau hyperphosphorylation in AD [23], as well as in the amyloidogenic processing of the amyloid precursor protein (APP) that generates Aβ_1–42_ [21].

In physiological conditions, the Ras/MAPK axis controls the dendritic spine formation. However, in AD, Aβ accumulation results in a dendritic spine and synapse loss, leading to cognitive decline. The molecule controlling this dendritic spine and synapse formation (and, therefore, learning and memory processes) is RasGRF1 GEF. When Aβ levels are reduced by a pharmacological treatment, mRNA and protein levels of RasGRF1 are increased, promoting dendritic spine formation in hippocampal primary neurons [24,25]. Considering that the RasGRF1/Ras/MAPK pathway is involved in the dendritic spine formation, benzothiazole amphiphiles such as hexa(ethylene glycol), derivative of benzothiazole aniline, have been effective in modulating this pathway and in promoting dendritic spine formation in human iPSC-derived neurons [26].

The Ras/MAPK pathway activity is also relevant in glia; for example, microglial phagocytosis requires its activation [27]. Nevertheless, the Ras/MAPK pathway can also be detrimental to cellular physiology, as it is involved in l-3,4 dihydroxyfenylalanine-(l-DOPA)-treated PD patient dyskinesia. In a PD-mouse model, l-DOPA induces dyskinesia by increasing the Ras GEF RasGRP1 protein levels. In contrast, l-DOPA-treated RasGRP1^-/-^ mice do not display dyskinesia [28]. Moreover, l-DOPA promotes the activation of extracellular signal-regulated kinase (ERK) and mTORC1 [29]. As RasGRP1 acts as a GEF for both Ras and Rheb (another Ras family GTPase), the authors suggested that in the presence of l-DOPA, RasGRP1 could form complexes with both GTPases. On one hand, it could associate with H-Ras to initiate ERK signaling; on the other hand, it could activate Rheb to signal via mTOR [29]. Hence, parallel activation of these two cascades could provoke the onset and progression of L-DOPA-induced dyskinesia [29].

The MAPK pathway is not the unique Ras-activated signaling pathway, as dendritic spine formation is achieved by a cooperation between the Ras/MAPK and Ras/PI3K/Akt pathways [30]. Although no evidence shows that Ras interaction with PI3K induces neurodegeneration, it is known that Ras/PI3K/Akt promotes neuron survival [31]. Furthermore, the Ras/PI3K/Akt pathway can also activate mTOR, leading to an increase in dendrite arborization, both in number and complexity, in the presence of the brain-derived neurotrophic factor (BDNF) [32]. Therefore, Ras/PI3K pathway abnormal functioning could be involved in pathologies that present dendritic spine loss such as AD. Furthermore, the PI3K/Akt/mTOR pathway has recently been described to be altered in diseases such as ischemic brain injury and neurodegenerative diseases such as AD, PD, and Huntington’s disease [33].

The Ras GTPases regulate the signaling pathway that controls cell–cell interactions through adherens junctions via its effector molecule afadin-6 (AF-6). Together with Parkin, AF-6 is involved in increasing mitophagy [28]. Basil and collaborators have demonstrated that human AF-6 overexpression in Parkin^-/-^ and PINK1^-/-^
*Drosophila melanogaster* models rescues the parkinsonian mitochondrial pathology, improving locomotor deficits and increasing fly survival [34]. Moreover, AF-6 overexpression in pathological mutant LRRK2^G2019S^ flies improved the parkinsonian phenotype. The authors demonstrated that AF-6 protected DA neuron dysfunction in flies [34]. All these data suggest that AF-6 activation could play a protective role in PD by regulating mitochondrial homeostasis [28,34].

Other Ras effector molecules potentially implicated in neurodegenerative processes such as AD and/or PD are RIN1 and Tiam1. Both of them control synaptic plasticity; RIN1 destabilizes synaptic connections by affecting the dendrite morphology and increasing the dendritic filopodial motility [35], whereas Tiam1 controls neurite outgrowth [36]. Tiam1 also controls the DA neuron differentiation [37]. Understanding the molecular mechanism that leads to this differentiation is crucial for the development of PD therapies based on DA neuron differentiation [37].

Ras effector molecules RIN1 and Tiam1 are characteristic GEFs. RIN1 is a Rab5A GEF, which regulates α-amino-3-hydroxy-5-methyl-4-isoxazolepropionic acid (AMPA) receptor endocytosis [35], while Tiam1 is a Rac1 GEF that controls cytoskeleton-related processes, such as neurite outgrowth [36] or Schwann cell migration [38].

In conclusion, Ras GTPases control the generation of toxic peptides such as Aβ_1–42_ and pTau via the MAPK pathway and microglial phagocytosis of fAβ. It also induces dendritic spine and synapse loss. All these are hallmarks of AD. In PD, Ras controls mitophagy and L-DOPA-induced dyskinesia (Figure 2A).

### 2.2. Rap GTPase

The Rap GTPases are in a subfamily (Rap1 and Rap2) of the Ras family of GTPases that regulate processes such as neuron migration and maturation as well as plasticity of dendritic spine and synapses [39]. The Rap GTPases are activated by GEFs such as Epac1, Epac2, C3G, RapGEF1, PDZ-GEF1, RA-GEF1, or RAPGEF2.

#### 2.2.1. Rap/MAPK/ERK

Like Ras, Rap functions upstream of the MAPK pathway. The Ras and Rap GTPases have a highly conserved effector domain in common [40], and they share some effector molecules. Whereas Rap1 activates the Raf/MEK/ERK pathway [41], Rap2 activates the c-Jun N-terminal kinase (JNK) pathway [42].

In dendrites, the Epac2-dependent Rap activation results in an increase in phosphorylated B-Raf (p-B-Raf) [43]. Treatment of primary pyramidal neurons with the cAMP analogue that activates Epac, known as 8-(4-Chlorophenylthio)-2′-O-methyladenosine 3′,5′-cyclic monophosphate monosodium hydrate (8-CPT-cAMP), provokes dendritic spine retraction [43]. Furthermore, the Epac/Rap axis mediates synapse structural destabilization, spine shrinkage, and AMPA receptor removal from spines [43].

Mutations in the *EPAC2* gene have been linked to autism [44]. In HEK293, Epac2^V646F^ reduced Rap activity and p-B-Raf levels in dendrites. Conversely, Epac2^T809S^ increased Rap activity and p-B-Raf levels. Moreover, both Epac2^V646F^ and Epac2^T809S^ also alter spine morphology [43].

Rap1 also controls neuronal activity via ERK. Phosphodiesterase 6δ (PDE6δ) interacts with Rap1 to control MAPK subcellular localization [45]. Thus, PDE6δ functions as a regulator of Rap1 by regulating the MAPK pools that will activate the GTPase. PDE6δ also regulates Rap1 recycling from the endomembrane to the plasma membrane [45]. Blockade of the PDE6δ/Rap1 interaction increases the presence and activity of Rap1 in the endomembrane, favoring the reduction of neuronal activity. In fact, the Rap1/ERK axis in the plasma membrane phosphorylates and inactivates potassium channels responsible for action potentials, resulting in neuronal hyperactivity. In contrast, the Rap1/ERK complex suppresses calcium influx through voltage-gated calcium channels (VGCC) and increases GABA_B_ receptor activity in the endomembrane. Therefore, this regulatory nexus could be a therapeutic target for treating diseases with neuronal hyperactivity [45], such as the early stages of AD. AD is characterized by an increase in intracellular calcium levels, which culminates in elevated excitability and, consequently, in neuronal death. Of note, PDE6δ/Rap1 inhibition has proven to partially rescue normal phenotype in AD [45].

Taking into account that the MAPK pathway is also involved in Tau hyperphosphorylation, inhibition of PDE6δ/Rap1 interaction in the mice models of AD hAPP*PS1 and hTAU reduced ERK cascade activation and Tau phosphorylation in different residues in vivo, which resulted in the rescue of behavioral deficits [45].

#### 2.2.2. Rap2/JNK

Rap2 controls JNK activity, one of the MAPK signaling nodes. JNK family is formed by three members, JNK 1–3. While JNK1 and 2 are ubiquitously expressed, JNK3 is mainly expressed in neurons. JNK can directly control apoptosis regulating protein Bcl-2 and c-Jun. Subsequently, JNK was linked to neurodegenerative diseases [46]. For instance, post-mortem AD brains displayed an increase in phosphorylated JNK [47].

Rap2/JNK is also involved in dendritic spine loss when activated by PDZ-GEF1. 3xTg-AD mice and human post-mortem AD brain samples showed increased protein levels of this GEF. Moreover, the treatment of cultured hippocampal neurons with Aβ oligomers increased PDZ-GEF1 protein levels, which subsequently activated Rap2. In AD, PDZ-GEF1 activates the JNK pathway culminating in a dendritic spine loss and cognitive decline [48].

#### 2.2.3. Rap1/AF-6

Like MAPK, Ras and Rap conserved sequence homology determines their sharing of effector molecule AF-6. Activated Rap1 recruits AF-6 to the plasma membrane, inducing spine neck elongation, as well as regulating AMPA receptor content in the spines [49]. Furthermore, it is a key molecule in spine formation and synaptogenesis [50].

#### 2.2.4. LRRK2/EPAC-1/Rap1 Pathway

Rap1 controls cell adhesion, polarization, and directional migration; processes that are best characterized in macrophages and microglia. In macrophages, the *Rap guanine nucleotide exchange factor 3* (*Rapgef3*) gene expression, which codifies for Epac1, is controlled by LRRK2 [51]. LRRK2 is a multifunctional protein. One of its activities, as a serine/threonine kinase, is the phosphorylation—a wide range of proteins involved in neuronal plasticity, autophagy, and vesicular trafficking [52,53,54]. Moreover, LRRK2 also regulates Rab family GTPases by phosphorylation [55] and it phosphorylates AD-related APP [54]. Apart from having a kinase activity domain, LRRK2 also presents a GTPase activity domain. The effect of the PD-related pathological mutation (LRRK2G2019S) in the kinase activity has been extensively studied, and it is known that LRRK2 mutations result in increased kinase activity that induces a neurotoxic effect [56]. In contrast, little attention has been paid to the effects of this mutation on the GTPase activity when it is known that the GTPase domain plays an important role in the regulation of the kinase activity and its neurotoxic effects [57].

A transcriptomics study showed that LRRK2 reduces Epac1 expression in macrophages [51]. Thus, macrophage migration regulated by LRRK2/Epac1/Rap1 might be one of the possible dysregulated pathways in PD [51]. Although this axis still needs to be characterized in microglia, Levy and collaborators suggest that the suppression of Epac1/Rap1 by PD-associated LRRK2 could reduce the ability of microglia to migrate to sites of neuronal cell damage in PD [51].

#### 2.2.5. Epac/Rap1/APP Processing

The Epac/Rap1 pathway may present a dual role. On the one hand, it regulates the neuroprotective cleavage of APP. In fact, α cleavage of APP is induced via the cAMP/Epac/Rap1STEF/Rac1 pathway activated by the serotonin 5-HT4 receptor [6]. Rap1 specifically associates with the TSS region of GEF STEF, which results in the activation of Rac1 GTPase of the Rho family [6]. This culminates in the secretion of soluble α APP (sAPPα), which has neuroprotective properties and promotes memory improvement.

On the other hand, Epac/Rap1 can regulate the toxic cleavage of APP. In the presence of Aβ or hypoxia, Epac/Rap1 is involved in miR-124 inhibition [58]. miR-124 regulates BACE1 secretase mRNA expression and protein levels, which is the enzyme responsible for the β cleavage of APP. Thus, in the presence of Aβ or hypoxia, miR-124 inhibition provokes an increase in BACE1 levels and, therefore, in the production of toxic Aβ [58]. Additionally, it is important to highlight that AD patients express low levels of miR-124 [59].

In a nutshell, Rap regulates both the amyloidogenic and non-amyloidogenic processing of APP and is also responsible for neuronal hyperactivity and dendritic spine loss (Figure 2B).

### 2.3. Rheb GTPase

Rheb GTPase, another member of the Ras family, is involved in cell cycle regulation, cell growth control [60], autophagy [61], and apoptosis by controlling the interaction of FKBP38 with proteins Bcl-2 and Bcl-XL [62]. Although it shares some domains with other Ras GTPases, such as the effector and GTP-binding domains or the motif for being prenylated, Rheb regulation is unique amongst the members of this family. The TSC1/TSC2 GAP complex regulates the Rheb activation/deactivation cycle. Thus, to activate Rheb, the PI3K/Akt/PKB axis phosphorylates TSC2, and in this fashion, inhibits the TSC1/TSC2 GAP complex [63].

Some evidence suggest that Rheb activation potentiates neurodegeneration via the GTPase association to its main effector molecule mTOR. Post-mortem AD brain samples showed reduced TSC2 protein levels, which implicates a continuous active state of Rheb; this suggests that Rheb activation could play a role in the disease [64]. Another study showed that in post-mortem AD and PD with dementia (PD/DLB) samples, the detected TSC2 was hyperphosphorylated, probably via Akt [65]. Because this phosphorylation resulted in GAP inhibition and consequent Rheb activation, this study suggested that this mechanism could trigger neuronal death in various neurodegenerative pathologies such as AD and/or PD [65]. Moreover, as previously explained, simultaneous signaling by the RasGRP1/Rheb/mTORC1 and RasGRP1/H-Ras/ERK axes could underlie the dyskinesia suffered by PD patients who have been treated with L-DOPA for a prolonged time [29]. mTOR has also been shown to induce Tau phosphorylation and its pharmacological inhibition alleviated the pathology and the behavior deficits in human Tau overexpressing mice. Presently, this data suggests that mTOR could be a therapeutic target for tauopathies [66].

Nevertheless, Rheb/mTOR implication in neurodegeneration is controversial. Other studies maintain that Rheb activation, and, therefore, mTOR activation, controls cell survival and protects from neurotoxicity in adult brain [67]. Accordingly, a constitutively active form of Rheb improved the PD phenotype by inducing axonal growth in DA neurons via Akt/mTOR [68]. Moreover, in the AD mouse model 5XFAD, constitutively active Rheb resulted in increased levels of neurotrophic molecules such as BDNF and ciliary neurotrophic factor (CNTF), and their respective receptors tropomyosin receptor kinase B (TrkB) and CNTF receptor α subunit (CNTFRα) [69]. Additionally, a constitutively active form of Rheb inhibited Aβ production and accumulation in the hippocampus of these mice and prevented cognitive impairments [67]. Hence, also Rheb represents a potential therapeutic target for neurodegenerative diseases [70].

Rheb can also modulate signaling cascades independently of mTOR. Recently, Rheb has been described to bind to BACE1 secretase, this being a novel GTPase effector molecule [71]. Rheb-GTP binds to BACE1 and promotes proteasomal and lysosomal degradation of this secretase in HEK293 [71]. In murine primary neuronal cultures, Rheb reduced BACE1 levels, as well as its activity, resulting in a decrease in sAPPβ, APP-CTFβ, Aβ_1–40_ y Aβ_1–42_ species. Shahani and collaborators also demonstrated that post-mortem AD samples showed reduced Rheb protein levels. Taking into account that BACE1 levels are increased in AD, Shahani et al. suggested that low Rheb levels could explain the increased levels of BACE1 and the consequent Aβ generation [71]. Likewise, low Rheb levels are associated with memory loss and, therefore, it could be a therapeutic target for pathologies characterized by memory problems such as AD [72].

To sum up, Rheb controls the production of Aβ peptides and it improves the PD phenotype by inducing axonal growth (Figure 2C).

## 3. Small GTPASES of the Rho Family

GTPases of the Rho family are specialized in the regulation of actin cytoskeleton dynamics, although they also control cell cycle progression or transcriptional regulation [3]. The best-characterized GTPases of this family, from a structural and a functional point of view, are RhoA, Rac1, and Cdc42.

The GEFs that control Rho GTPases activation are divided into two families: Dbl and DOCK [4,69,73]. In the Dbl family, we can find αPIX, Vav1, Tiam-1, Tiam-2/STEF, RasGRF, Sos1, Dbl, RhoGAP, p115RhoGEF, and Trio. In the DOCK family, its members are divided into four subfamilies depending on the sequence homology and the substrate specificity: DOCK-A (DOCK2, 5, and 180), -B (DOCK3 and 4), -C (DOCK6, 7 and 8), and -D (DOCK9, 10, and 11).

Rho GTPases, as well as their regulators and effector molecules, play an important role in neurodegenerative processes, such as regulating APP processing or dendritic spine loss in AD [17] and oxidative stress and neuroinflammation in PD [18,74]. Furthermore, Rho GTPases are also implicated in other diseases related to the nervous system [12]. For instance, they are involved in amyotrophic lateral sclerosis (ALS) by regulating motor neuron survival and they control Huntingtin protein accumulation in Huntington’s disease [12].

### 3.1. RhoA

RhoA is the GTPase that gives the name to this family of GTPases, where other closely related isoforms such as RhoB and RhoC can be found [2]. RhoA controls the formation of actomyosin contractile fibers, also known as stress fibers.

Changes in RhoA subcellular localization have been linked to neurodegenerative processes occurring in AD [75]. APP_Swe_ Tg2576 mice express reduced levels of RhoA in synaptic ends and increased levels in dystrophic neurites [75]. RhoA has also been connected to PD pathology [76,77]. The intracellular signaling network controlled by RhoA is complex, being the best characterized effector molecules Rho-kinase (ROCK) and the formin subfamily Diaphanous (Dia).

#### 3.1.1. RhoA/ROCK

RhoA controls actin cytoskeleton reorganization through phosphorylation carried out by the ROCK kinases family resulting in the activation of proteins such as myosin light chain (MLC) or LIM kinase (LIMK), which in turn phosphorylates and activates cofilin.

ROCK kinases have been associated with AD, as they can phosphorylate Tau protein and mediate NFT formation [78]. Studies in post-mortem AD samples showed increased ROCK expression levels [79]. Furthermore, ROCK promotes an increase in secreted Aβ_1–40_ levels and, in turn, increased Aβ levels potentiate ROCK activity, as well as its effector LIMK activity [79]. Henderson and collaborators proposed that a positive feedback loop between Aβ/RhoA/ROCK/Aβ could be favoring AD progression [79].

The RhoA/ROCK axis has also been linked to AD pathology as it is involved in the retraction of dendritic spines. Aβ peptides induce an increase in actin contractility via Pyk2/RhoGAPGraf1/RhoA-regulated ROCK2 kinase culminating in dendritic spine retraction [80].

In microglia, Aβ peptide requires the RhoA/ROCK axis to induce different responses such as chemotaxis, cytotoxicity, and inflammatory response [81]. Thus, the RhoA/ROCK signaling in microglia could be favoring AD neurodegeneration. Based on this hypothesis, Scheiblich and Bicker defended the therapeutic potential of RhoA/ROCK inhibitors for treating excessive inflammation and neurodegeneration in CNS microglia [82].

RhoA/ROCK has also been implicated in DA neuron degeneration in PD. Both RhoA and ROCK inhibition presented a therapeutic effect in PD mouse models [76,77]. In addition, RhoA inhibition by C3 transferase resulted in a reduction in α-syn (in mRNA and protein levels) in the MN9D neuronal cell line; this suggests RhoA as a potential therapeutic target for synucleopathies [83]. Although Zhou et al. did not describe the molecular events that led to this α-syn reduction, they observed that RhoA inhibition provoked a reduction of the SRF transcription factor in the nucleus, as well as cytosolic retention of MKL-1 (co-transactivator of SRF that translocates to the nucleus to bind SRF and, therefore, regulates transcription). Hence, RhoA inhibitory effects could be due to a reduction in transcription [83].

#### 3.1.2. RhoA/NOX

RhoA can also activate the holoenzyme NADPH oxidase (NOX) to generate superoxide ions [84]. In the presence of fibrillar Aβ in microglia, RhoA/NOX is activated promoting the production of superoxide species [84]. RhoA controls the superoxide production by regulating the phosphorylation of the p47^PHOX^ subunit of the holoenzyme [84]. Microglial NOX is considered an important source of oxidative stress that results in AD neuronal cell death [85].

In PD, microglial activation via angiotensin receptor AT1 induces activation of the RhoA/ROCK/NOX2 axis and subsequent superoxide generation [86]. α-syn 29–40, a α-syn specific peptide, has been described to activate superoxide generation in microglia, which causes DA neuronal damage [87]. Later, it was described that extracellular α-syn binds to CD11b integrin in microglia and controls NOX2 activation via a RhoA-dependent pathway [88].

Therefore, RhoA participates in the accumulation of toxic peptides in AD. Moreover, it induces toxic responses such as microglial cytotoxicity and inflammatory response in AD, and neurodegeneration in PD (Figure 3A).

### 3.2. Rac GTPase

The Rac GTPases belong to a subfamily of Rho GTPases constituted by four members—Rac1, Rac2, Rac3, and RhoG—where Rac1 is the best studied member. These GTPases control cytoskeleton organization, gene transcription, and proliferation [3]. Together with Ras and Rho, Rac is considered a protooncogene, and mutations in these GTPases confer them a constitutive activity in various types of tumors [11]. Apart from participating in tumoral processes, Rac1 plays a key role in development and neuronal survival as well as in neurodegeneration [12].

In AD, Rac1 has been described to increase Aβ levels by transcriptional regulation, as it increases APP expression by controlling the −233 to −41 bp positions in the promotor of the APP gene [89]. Recently, Borin et al. have reported that a constitutively active form of Rac1 increases APP processing, culminating in a higher content of Aβ_1–42_ peptides [90]. Thus, Rac1-induced increase in Aβ levels could be due to an increase in APP expression as well as in its processing. Conversely, Rac1 is also involved in the reduction of Aβ_1–42_ levels as it controls the microglial phagocytosis of Aβ_1–42_ peptides via its effector molecule WAVE [91].

#### 3.2.1. Rac1/Toxic Peptide Accumulation

Like RhoA/ROCK, Rac1 also controls Tau phosphorylation in the early stages of AD. Rac1 induces the translocation of SET from the nucleus to the cytoplasm [90]. SET, as an inhibitor of Tau phosphatase, blocks Tau dephosphorylation.

In PD, α-syn accumulation is toxic and is responsible for PD symptomatology. Rac1 reduces α-syn accumulation in *Caenorhabditis elegans*, in the BE(2) M17 human neuroblastoma cell line and DA neuron-like cells derived from iPSCs from PD patients [92]. This can be viewed as a cell survival mechanism promoted by Rac1 [92].

#### 3.2.2. Rac1/PAK

PAK (p21 activated kinase) is a family of serine/threonine kinases that function as Rac and Cdc42 GTPases effector molecules, and they are implicated in the signal transduction emanating from integrins and tyrosine kinase receptors. Furthermore, PAK has been related to neurodegenerative diseases [93]. In a healthy brain, PAK is involved in the regulation of neuron viability, synapse morphology and functionality, gene transcription in neurons, microglial motility, microglial immune response, and astrogliosis [94]. In contrast, PAK protein and activity levels are elevated in neurodegenerative disorders, resulting in a decrease in neuronal cell viability and, therefore, a potentiated neuronal degeneration in diseases such as AD and PD [94].

Nevertheless, the role of this over-activated Rac1/PAK axis in dendritic spine loss is controversial, as some authors understand that Rac/PAK helps the formation of dendritic spines [95]. Borin et al. claimed that Rac might present a dual role depending on the stage of AD disease [90]. Whereas Rac1 is over-activated in early stages of the disease, in later stages, Rac1 protein levels decrease, which results in a dendritic spine loss [90]. Altogether, the role of Rac1/PAK in AD is complex and still needs further studies, considering different factors such as the model used for the study and the stage of the disease.

In PD, wild type LRRK2 binds strongly to Rac and increases its activity. This can be seen as an increase in the binding of Rac1 to PAK [96]. The LRRK2^G2019S^ mutant phenotype presents neurite retraction, but in SH-SY5Y cells when Rac1 was co-expressed with LRRK2 ^G2019S^, Rac1 was able to rescue this retraction [96]. Thus, Chan et al. postulated that pathological mutations in LRRK2 could attenuate Rac1 activation, resulting in an actin filament disassembly and, consequently, neurite retraction [96].

#### 3.2.3. PI3K/PDK/nPKC/Tiam-1/Rac1/Neuronal Cell Death

Rac1 is implicated in Aβ_1–42_-induced neuronal cell death. In the SN4741 neuronal cell line, Aβ_1–42_ requires the PI3K/PDK/nPKC axis to promote Tiam-1 phosphorylation and, subsequently, activate Rac1 GTPase culminating in the apoptosis of SN4741 cells, primary embryonic cortical neurons from rats, as well as in neuronal organotypic cultures of the hippocampus and the entorhinal cortex [97]. Manterola et al. highlighted the central role of Rac1 in transducing the Aβ_1–42_-mediated toxic signaling [97]. Although the signaling cascade inducing neuronal cell death downstream of Rac1 is still unknown, the authors suggested that one of the mechanisms could be the Rac1 activation of NOX and the ensuing generation of reactive oxygen species (ROS).

#### 3.2.4. Rac1/NOX

NOX is also an effector molecule of Rac1. Rac1/NOX axis is also involved in neurodegenerative processes [98,99].

In murine primary microglial cell cultures, fibrillary Aβ (fAβ) activates Rac1/NOX, generating oxidative stress that affects neuronal viability [98]. Moreover, fAβ binds to different microglial receptors, inducing the activation of Lyn and Syk kinases. These kinases are responsible for phosphorylating Vav, a Rac1 GEF, resulting in Rac1 activation and, subsequently, NOX [98]. The fAβ/Lyn/Syk/Vav/Rac1 axis, apart from activating NOX, also induces actin cytoskeleton organization that promotes microglial phagocytosis [98]. In primary astrocytic cultures from mice, Aβ_1–42_ induces astrogliosis [99], which is an astrocytic hypertrophy that compromises neuronal cell viability. Aβ_1–42_ induced this astrogliosis by NOX activation via the PI3K/cPKC/Rac1/NOX axis [99].

Rac1 activation and oxidative stress generated by NOX are also involved in PD [74]. Rac1 inhibition and NOX1 silencing resulted in a reduction of DNA damage and DA neuronal degeneration in the N27 cell line and DA neurons in the *substantia nigra* of rats [74].

#### 3.2.5. Rac1/JNK

JNK signaling pathway, together with the p38 and ERK MAPK pathways, are activated in different stages of AD. JNK activates apoptotic processes [46]. Moreover, it is also able to phosphorylate Tau protein, favoring the NFT formation, and APP activating its processing [46]. All these processes participate in AD progression [46]. On the other hand, JNK could also be a therapeutic target for PD [100]. When studying the Rac1/NOX1 axis in DA neuronal cell death, Choi et al. reported that Rac1 inhibition and NOX1 silencing resulted in a reduction of the apoptotic marker c-Jun, which is phosphorylated by JNK [74]. These results suggest that lower levels of phosphorylated c-Jun could be indicative of a good prognosis for PD.

In conclusion, Rac participates in the accumulation of the three toxic peptides Aβ, pTau, and α-syn in AD and PD, respectively. It is also responsible for inducing toxic effects such as apoptosis or ROS generation in neurons, phagocytic pathways activation in microglia, and astrogliosis control in astrocytes (Figure 3B).

### 3.3. Cdc42 GTPase

Cdc42 promotes neurite formation, axon growth and ramification, and spine formation in the nervous system [101]. Moreover, it plays an important role in neurogenesis, as it participates in the progenitor cell formation and also in their differentiation into neurons [102]. Cdc42 regulation of these functions requires its activation by different GEFs such as Dbs, intersectin (ITSN), Prex1, Tiam1, Vav1–3, and Fgd5 [4]. ITSN-1 was linked to neurodegenerative diseases, as its mRNA expression is increased in AD brain samples [103]. Moreover, ITSN overexpression alters Cdc42-mediated endocytosis.

Considering that anomalies in the endocytic pathway are present in AD and that ITSN may contribute to AD pathology, ITSN-Cdc42 interaction could be a therapeutic target to treat this disease [17].

It has been demonstrated using post-mortem brain samples that Cdc42 expression is elevated in the prefrontal cortex of AD patients [104]. Gene expression database analyses have shown that PD patients have reduced Cdc42 mRNA levels [93]. Together, this data indicates that Cdc42 could play an important role in neurodegeneration. Of note, the neural Wiskott–Aldrich Syndrome protein (N-WASP) is a Cdc42-specific effector molecule, while PAK, PI5K, or formins are Cdc42-effector molecules shared with Rac. All of them are involved in actin cytoskeleton reorganization. It is necessary to understand all these downstream cascades in order to identify potential targets for neurodegeneration.

#### 3.3.1. Cdc42/ PAK

This protein complex controls neurite growth in the same way as Rac1. It also controls filopodia formation to regulate axon guidance, the process by which neurons extend their axons to reach their targets [105]. Cdc42/PAK is also necessary for dendritic spine formation [106]. PAK stimulates LIMK, which inhibits actin microfilament depolymerization. The dysregulation of this axis has been linked to AD [107].

The DA neuron arborization and process formation also depends on the Rho family of GTPases. Constitutive activation of Cdc42, Rho, and Rac when adding cytotoxic necrotizing factor 1 (CNF1), resulted in hypertrophic DA neurons with an increased length of the ramifications in cultures from rat *substantia nigra*. In vivo, CNF1 treatment in 6-OHDA-treated mice improved motor asymmetries. Thus, Rho GTPases display a neurorestorative potential for PD [108].

#### 3.3.2. Cdc42/N-WASP

The WASP protein family functions as actin nucleation promoting factors, by regulating the Arp2/3 complex activity, which is responsible for the formation of actin filaments [109]. These filaments control processes such as lamellipodium and filopodium formation, endocytosis, phagocytosis and vesicle generation from Golgi, endoplasmic reticulum, or endolysosomal traffic [109]. Among the WASP family members, Cdc42 specifically increases N-WASP activity towards Arp2/3. N-WASP has a low basal activity, which is dramatically increased by Cdc42. It is important to highlight that the AD patients samples demonstrated increased N-WASP protein levels, suggesting that this protein could be involved in AD neurodegeneration [110].

The ITSN/Cdc42/N-WASP pathway is involved in spine morphogenesis [111] since Ephrin type-B receptor 2 (EphB2) associates with ITSN/Cdc42 favoring spine morphogenesis [111].

#### 3.3.3. Cdc42/GSK3

Glycogen synthase kinase-3 (GSK3) is a constitutively active serine/threonine kinase implicated in the regulation of various processes such as apoptosis, cell survival, protein translation, or cell cycle progression. Some of these functions are associated with neurodegeneration [112].

There are two GSK3 isoforms, α and β, codified by two different genes. In the CNS, GSK3-β is the most abundant and its levels increase with age [113]. The activity of this isoform has been associated with memory, neurogenesis, synaptic plasticity, long-term potentiation (LTP), and inflammation [112].

Samples from AD patients show increased GSK3-β protein levels [114]. In addition, the dysregulation of this kinase both in in vitro and in vivo AD models affects Aβ and Tau metabolism [112]. In AD, increased neuronal isoprenoid levels induce Cdc42 prenylation, promoting Tau phosphorylation and the consequent NFT formation [115]. Cdc42 activates GSK3β, which is responsible for Tau phosphorylation. Furthermore, Aβ peptides induce an increase in GSK3 activity and Tau hyperphosphorylation, although it has not been described whether Cdc42 mediates this Aβ-dependent GSK3 activation.

#### 3.3.4. Cdc42/Endocytosis

As previously mentioned, ITSN overexpression alters endocytosis via Cdc42. Taking into account that AD patients present endocytic traffic anomalies, ITSN could be a contributor to AD pathology, and the ITSN–Cdc42 interaction could be a therapeutic target to treat the disease [17].

Recently, Cdc42 and RhoA were identified as Aβ_1–42_ endocytosis regulators. Constitutively active forms of Cdc42 and RhoA promoted a reduction in Aβ_1–42_ endocytosis in SH-SY5Y cells [116]. It suggests that the Rho family of GTPases controls Aβ_1–42_ internalization and that the endocytic pathway could be a target to reduce Aβ_1–42_.

Conclusively, Cdc42 controls GSK3-mediated pTau and NFT accumulation, and it also controls dendritic spine formation and Aβ endocytosis. In PD, together with Cdc42, the other Rho family GTPases regulate neuronal hypertrophy (Figure 3C).

## 4. Conclusions and Future Perspectives

The role of small GTPases of the Ras superfamily has been overlooked for a long time in neurodegenerative diseases. Recently, they have risen as key players in the pathogenesis of many cerebral diseases, as well as their regulatory and effector molecules.

In a broad concept, Rho GTPases in physiological conditions are responsible for the actin cytoskeleton remodeling, and their alteration may produce abnormalities such as spine loss in neurons. Moreover, considering the high number of effector molecules of Rho GTPases, their up- or downregulation may lead to pathological events such as neuronal cell death. Regarding the Ras family, it induces neurodegeneration via the MAPK. There are many reasons to consider small GTPases as therapeutic targets. However, due to their dual roles in neurodegenerative diseases, the treatment based on GTPases is somehow complex. Nevertheless, the activity of the small GTPases of the Ras superfamily could be modulated by adding reagents that enhance or inhibit their activity, depending on their pathological state of activation. For instance, astrocytic migration regulated by Rho GTPases is a hallmark of AD, and ibuprofen and its derivatives regulate cytoskeleton dynamics by suppressing Rho GTPases activities. Thus, ibuprofen may be helpful in preventing this disease [117]. The advantage of this treatment would be that ibuprofen is already available for use in humans, so it would be able to skip some steps of clinical trials. Furthermore, thiopurine, an active substance for inflammatory bowel disease that suppress Rac1 activation, resulted in a lower rate of AD in U.S. veterans with inflammatory bowel disease, and for each additional year of thiopurine exposure, the risk of AD is reduced by 8.3% [118]. The association between Rac1 and the AD neurodegeneration process opens up the possibility that Rac1 might be triggering early neurodegenerative responses at the onset of AD. Regarding the bacterial toxin cytotoxic necrotizing factor 1 (CNF1), its activity is the opposite acting as an activator of Rho GTPases, and its administration has been shown to correct deficits in reversal learning in an animal model of AD, which might be beneficial for the treatment of AD patients [119]. In the case of PD, the continuous activation of Rac1, RhoA, and Cdc42 through the administration of CNF1 has demonstrated to improve the phenotype and to restore the degraded dopaminergic tissue in an animal model of PD [108]. Therefore, the activation of Rho GTPases may sometimes be beneficial, whereas at other times, their inhibition can ameliorate the disease.

Another strategy aims at the PTMs that allow an activation of GTPases, rather than directly activating or inhibiting the GTPase itself. Statins inhibit the pathway that is required for post-translationally modifying the GTPases [120]. This PTM allows them to be anchored to the plasma membrane, where they are activated. Therefore, the inhibition of this PTM could permit the regulation of GTPases activation and the modulation of the downstream signal transduction cascades. This strategy could be extended to many members of the small GTPases of the Ras GTPases superfamily. More recently, a statin called lovastatin was shown to promote myelin repair by inhibiting the activities of Rho family GTPases in glial cells [120].

The regulation of molecules that activate or inhibit GTPases is another approach that could be used for the treatment of neurodegenerative diseases. In this regard, GEFs inhibitors such as NSC23766 could be used where the GTPases are over-activated, and conversely, GAPs activators in diseases where GTPases are downregulated. Apart from regulating GEFs and GAPs, another option would be regulating GDIs, which function by maintaining the GTPase in the inactive form. Finally, the use of peptides that interfere with the interaction between the GTPases and their effector molecules is gaining research ground. The advantage of this methodology is that it can specifically inhibit the interaction between the GTPase with its effector molecule, without altering the interaction with other molecules. Future research will allow us to characterize the effectors and signaling molecules that participate in these signal transduction pathways.

Given the potential of Ras superfamily GTPases, Aguilar et al. have reviewed the use of Rho GTPases as therapeutic targets for AD. RhoA and Rac1 inhibitors, as well as inhibitors of their effectors and regulators, have been proven to be effective in reducing Aβ_1–42_-mediated toxicity in vitro [17]. However, before considering a GTPase as therapeutic target, it is important to describe the whole signaling cascade controlled by each GTPase in each specific pathological condition. In this regard, for instance, Manterola et al. disseminated that PI3K/PDK/nPKC/Tiam-1/Rac1 precisely leads to neuronal death mediated by Aβ_1–42_ [97]. Another significant example is the description of the axis RasGRF1/Ras/MAPK that controls the dendritic spine formation. Based on this, benothiazole aniline derivatives have been able to modulate this pathway in order to promote dendritic spine formation [26]. Moreover, Pyk2/RhoGAPGraf1/RhoA/ROCK2 cascade is responsible for the dendritic spine retraction in the presence of Aβ peptides [80]. Thus, the description of the exact axis controlling each toxic response would help us to identify therapeutic targets. Considering this, the field advances in describing the precise signaling cascades controlled by Ras superfamily GTPases in neurodegeneration in order to detect potential therapeutic targets.

However, not only is it important to describe the signaling cascades controlled by Ras superfamily GTPases, but the location of the specific pool that is being activated is relevant too. In fact, Dumbacher et al. described the role of PDE6δ in controlling neuronal activity by regulating the location of active Rap1/MAPK pools [45]. This opens a therapeutic venue where the location of active small GTPases could be used as target for diseases such as AD.

Furthermore, the implication of glial cells should not be disregarded. The description of Ras and Rho GTPases controlling microglial phagocytosis and superoxide generation in AD and PD [27,86,87,88,98] and controlling astrogliosis [99] has placed glial cells in the focus of many researchers. Therefore, methods specifically targeting glial cells could be a promising therapeutic option.

Nevertheless, the field is not only focused on the search for therapies. In fact, most neurodegenerative diseases lack of an early detection. AD and PD prognosis, as well as other neurodegenerative diseases, are improved when they are detected early. Therefore, in the same way as in the field of oncology, a liquid biopsy-based early diagnosis would improve the outcome of neurodegenerative diseases such as AD [121]. Healthcare systems should detect early markers by liquid biopsies, which is a non-invasive, time- and cost-saving technique. Therefore, deep research is currently being done in order to find biomarkers for early detection of neurodegenerative diseases [121].

Some of these biomarkers could be related to cascades controlled by Ras superfamily GTPases. In the case of AD, biomarkers related to amyloidogenic pathways, neuroinflammation, axonal damage, synaptic dysfunction, and synaptic loss are being analyzed [121]. In this regard, promising blood-based biomarkers are Aβ_1–42_/Aβ_1–40_ ratio, BACE1 activity measurements, and Tau levels [122]. In the case of PD, α-syn and the epidermal growth factor are some examples of blood-based biomarkers that are being studied [123].

However, researchers looking for blood-based biomarkers for detecting neurodegenerative diseases have to face many challenges. First of all, as reviewed by Hampel et al., if the biomarker needs to be present in blood, it has to be able to cross the blood–brain barrier [122]. Another challenge is that the concentration of the biomarker in blood could be lower in comparison to the concentration in the cerebrospinal fluid; for instance, Aβ concentrations are 10-fold lower in plasma [122]. This means that a high sensitivity is required for the detection of biomarkers in blood.

Once blood-based biomarkers for the detection of neurodegenerative diseases are established, the healthcare system will require a deep structural and functional renovation to perform large-scale screenings [121]. Soon, a liquid biopsy-based diagnosis will likely become a reality in the field of neurodegenerative diseases [121].

## Figures and Tables

**Figure 1 ijms-21-06312-f001:**
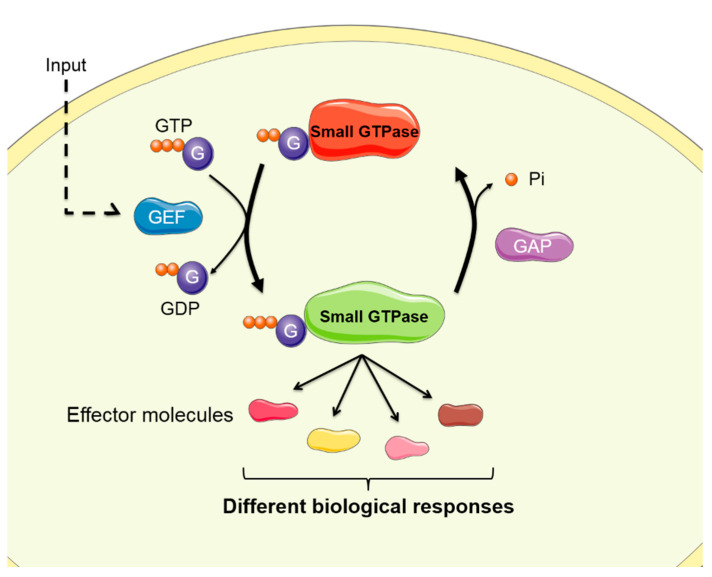
Activation/deactivation cycle of small guanosine triphosphatases (GTPases). The input receiving cells mediate GEF activation, facilitating the transition from the inactive configuration (guanosine-5′-diphosphate (GDP)-loaded, in red) to the active configuration (GTP-loaded, in green). In the active form, small GTPases interact with effector molecules in order to transduce signals that generate different cellular responses. On the other hand, GAPs promote the hydrolysis of GTP and the consequent inactivation of the GTPase. GEF: guanine nucleotide exchange factor. GAP: GTPase activating protein.

**Figure 2 ijms-21-06312-f002:**
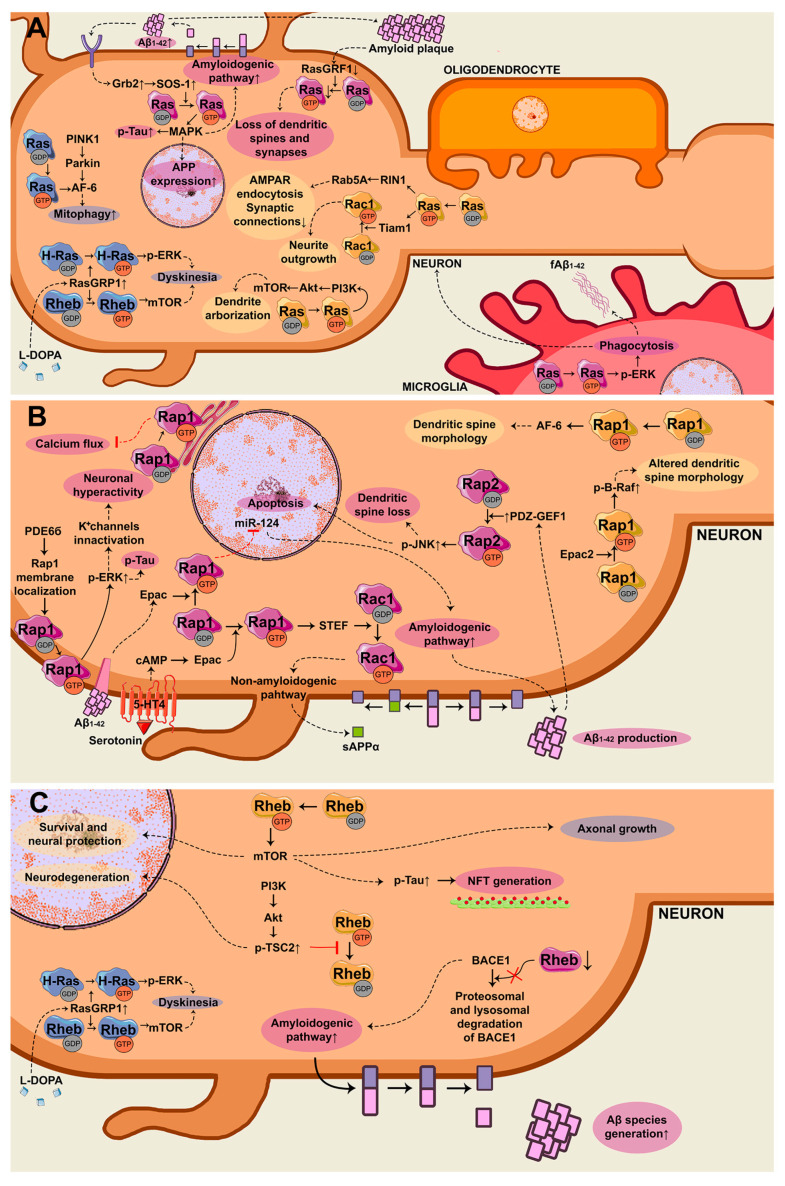
Scheme of the signaling pathways controlled by small GTPases of the Ras family, which are deregulated in general (yellow), in Alzheimer’s disease (AD) (purple), and Parkinson’s disease (PD) (blue) in neurodegenerative diseases. (**A**) Signaling pathways controlled by Ras GTPase in neurons and microglia. Growth factor receptor-bound protein 2 (Grb2)/SOS-1/Ras/ mitogen-activated protein kinase (MAPK), decreased RasGRF1/Ras, Ras/afadin-6 (AF-6), and Ras/RIN1 promote different responses that induce neurodegeneration. Ras/Tiam1 and Ras/PI3K in neurons, and Ras/extracellular signal-regulated kinase (ERK) in microglia, which could be altered in neurodegenerative diseases. (**B**) Signaling cascade controlled by Rap GTPases in neurons. Rap1/MAPK and PDZ-GEF1/Rap2/c-Jun N-terminal kinase (JNK) favor neurodegenerative processes. Rap1-induced miR-124 inhibition increases the production of toxic amyloid-β (Aβ). PDE6δ controls Rap1/MAPK pools that are driving to neurodegenerative responses. However, Rap1/STEF/Rac1 is inducing a protective phenotype. (**C**) Signaling axis controlled by Rheb GTPase. Rheb/mTOR can or cannot drive neurodegeneration depending of the input in neurons. Furthermore, reduced Rheb protein levels provoke a reduction in BACE1 degradation, which induces the generation of Aβ species.

**Figure 3 ijms-21-06312-f003:**
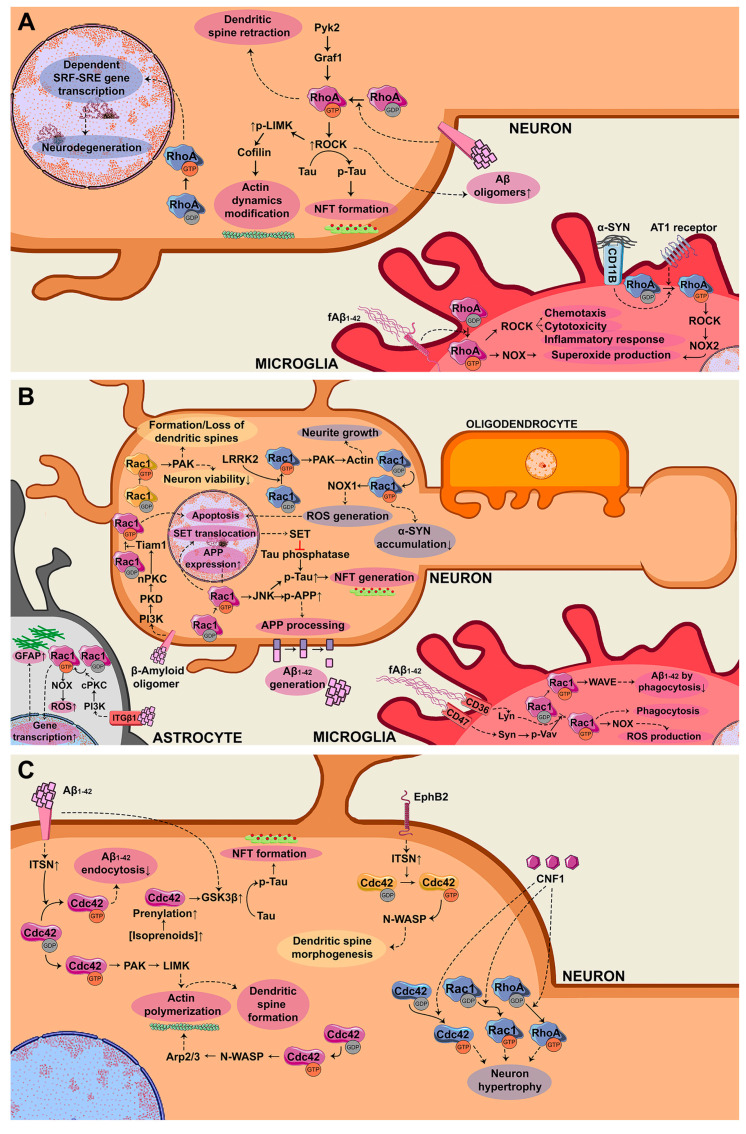
Scheme of the signaling pathways controlled by small GTPases of the Rho family, which are deregulated in general (yellow), in AD (purple), and PD (blue) in neurodegenerative diseases. (**A**) RhoA GTPase signaling pathways in neurons and microglia. RhoA/ROCK controls SRF–SRE gene transcription in neurons. In microglia, RhoA is activated via α-Synuclein (α-syn)/CD11B/RhoA and AT1/RhoA. (**B**) Signaling axis controlled by Rac1 GTPase in neurons, astrocytes, and microglia. In neurons, Tiam1/Rac1, Rac1/NOX1, and Rac1/JNK are inducing neurodegeneration, whereas the Rac1/PAK axis can drive neurodegeneration according to the input. Rac1 also decreases α-syn accumulation. In astrocytes, ITGβ1/PI3K/cPKC/Rac1 is leading to neurodegeneration, promoting astrogliosis by an increase of its gene expression targets and reactive oxygen species (ROS) generation. In microglia, the Vav/Rac1 axis is affecting neuronal viability by both, generating ROS and phagocytosis activation. On the other hand, Rac1/WAVE is decreasing Aβ_1–42_ levels by activating phagocytosis. (**C**) Signaling cascade controlled by Cdc42 GTPase in neurons. The axes intersectin (ITSN)/Cdc42 and Cdc42/GSK-3β, together with RhoA and Rac1, are driving to neurodegenerative responses.

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
