# Peer review of "Small GTPases of the Ras and Rho Families Switch on/off Signaling Pathways in Neurodegenerative Diseases"

_ijms, 2020, doi:10.3390/ijms21176312_

Round 1

Reviewer 1 Report

In this review article the authors summarize the current knowledge about the roles of small GTPases of the Ras and Rho families for neurodegenerative diseases. The literature is well covered, and the manuscript gives a good overview of the current literature. The illustrations support the text and illustrate the multiple roles of these small molecules in neuronal and glial cells.

The organization of the manuscript is systematic and the important aspects are covered starting from the role of the molecules in normal neuronal and glial cells to alterations of expression in neurodegenerative diseases and possible drug targets emerging from the current research. The strength of this review article is certainly its rather complete and extensive coverage of the relevant literature.

There are two weaknesses:
1) The inclusion of MS as a neurodegenerative disease means that may immunological data are included. This makes the whole article a bit heterogeneous because the diverse functions in immunology make the manuscript longer and somehow less focused. This reviewer would prefer to take this part out of the review and focus on the classical neurodegenerative diseases.
2) The review is rather descriptive, listing and enumerating all relevant published findings without commenting on their importance, novelty and relevance for neurodegenerative diseases. From this review, it is not possible to identify important and seminal papers in the field. As the small GTPases covered in the review are important for very basic cellular functions, it also remains open if changes of their expression or function in neurodegenerative diseases are really directly involved in the disease pathology or are simply reflecting the altered cellular metabolism and function in neurodegenerative diseases. If the authors would dare to give more guidance to the reader which of the covered articles are really advancing the field, this would greatly improve this manuscript.

Author Response

Dear Editor:

We received an electronic mail from you on August 24th 2020 with the opinion of two reviewers on our manuscript entitled Small GTPases of the Ras and Rho families switch on/off signaling pathways in neurodegenerative diseases” (Manuscript ID: ijms-894791; Type of manuscript: Review); submitted on July 25th, 2020.  The reviewers raised a number of concerns that you felt had to be addressed before considering the work fully acceptable for publication. With this letter, we are now enclosing a revised version of our manuscript in which we have addressed and/or corrected the issues pointed out by the reviewer.

We discuss below the criticisms given by the reviewers and how we have changed our work to solve them.

COMMENTS TO REVIEWER

REVIEWER 1:

This reviewer indicates that there are two weaknesses:

Point 1

1) The inclusion of MS as a neurodegenerative disease means that may immunological data are included. This makes the whole article a bit heterogeneous because the diverse functions in immunology make the manuscript longer and somehow less focused. This reviewer would prefer to take this part out of the review and focus on the classical neurodegenerative diseases.

Response point 1: We agree, we have taken MS part out of the review in order to focus on the classical neurodegenerative diseases. References numbers have been rearranged, as some references have been removed.

Point 2

2) The review is rather descriptive, listing and enumerating all relevant published findings without commenting on their importance, novelty and relevance for neurodegenerative diseases. From this review, it is not possible to identify important and seminal papers in the field. As the small GTPases covered in the review are important for very basic cellular functions, it also remains open if changes of their expression or function in neurodegenerative diseases are really directly involved in the disease pathology or are simply reflecting the altered cellular metabolism and function in neurodegenerative diseases. If the authors would dare to give more guidance to the reader which of the covered articles are really advancing the field, this would greatly improve this manuscript.

Response point 2: We agree, we have added a discussion in the end of the Conclusions and Future Perspectives part. Here, we highlight the papers that have been important in the field, as well as the direction in which the field is advancing.

Reviewer 2 Report

The paper by Sastre et al. is an interesting review addressing the involvement of small GTPases in neurodegeneration. The article is on 139 references mostly published in recent years.

Major issue:

List of abbreviations should be added to the text.

The quality of figures could be improved as these are to small and of insufficient resolution.

Minor issues:

Abstract section:

line 23, please add the phrase "and in some small GTPases also guanine nucleotide dissociaton inhibitors (GDIs).

line 82 is a repetition (see line 42)

line 56 should be "vice versa" not "viceversa"

line 139 is a repetition (see line 93)

line 165 should be "GTPases" not GTPASES"

line 170 should be "Ras GTPases" not "RASGTPases"

Some punctuation errors are to be corrected (dots and commas).

Author Response

Dear Editor:

We received an electronic mail from you on August 24th 2020 with the opinion of two reviewers on our manuscript entitled Small GTPases of the Ras and Rho families switch on/off signaling pathways in neurodegenerative diseases” (Manuscript ID: ijms-894791; Type of manuscript: Review); submitted on July 25th, 2020.  The reviewers raised a number of concerns that you felt had to be addressed before considering the work fully acceptable for publication. With this letter, we are now enclosing a revised version of our manuscript in which we have addressed and/or corrected the issues pointed out by the reviewer.

We discuss below the criticisms given by the reviewers and how we have changed our work to solve them.

COMMENTS TO REVIEWER

REVIEWER 2:

This reviewer indicates that there are two major issues:

Major issues

Point 1

List of abbreviations should be added to the text.

Response point 1: We agree, we have added a list of abbreviations used in the review at the end.

Point 2

The quality of figures could be improved as these are to small and of insufficient resolution.

Response point 2: We agree, we have increased the size of the texts in the figures, as well as the resolution of the figures.

Minor issues:

Abstract section:

line 23, please add the phrase "and in some small GTPases also guanine nucleotide dissociaton inhibitors (GDIs).

line 82 is a repetition (see line 42)

line 56 should be "vice versa" not "viceversa"

line 139 is a repetition (see line 93)

line 165 should be "GTPases" not GTPASES"

line 170 should be "Ras GTPases" not "RASGTPases"

Some punctuation errors are to be corrected (dots and commas).

Response minor issues: We agree, we have done all the corrections detected by the reviewer 2.